# *Ptgds* downregulation protect vestibular hair cells from aminoglycoside-induced vestibulotoxicity

Chen Chen[1☯], Zhimin Zhao[2☯], Jinghong Han[3], Yue Zhang[3], Guohui Nie[3,4]*

1 Collaborative Training Base of Shenzhen Second People's Hospital, Hengyang Medical School, University of South China, Hengyang, Hunan, China, 2 Department of Otolaryngology, Huazhong University of Science and Technology Union Shenzhen Hospital, Shenzhen, China, 3 Shenzhen Key Laboratory of nanozymes and Translational Cancer Research, Department of Otolaryngology, Shenzhen Institute of Translational Medicine, The First Affiliated Hospital of Shenzhen University, Shenzhen Second People's Hospital, Shenzhen, China, 4 State Key Laboratory of Chemical Oncogenomics, Guangdong Provincial Key Laboratory of Chemical Genomics, Peking University Shenzhen Graduate School, Shenzhen, China

☯ These authors contributed equally to this work.

* nieguohui@email.szu.edu.cn

## Abstract

The clinical use of aminoglycosides often results in injury to vestibular hair cells and subsequent vestibular dysfunction. Thus, clarifying the targets and mechanisms underlying aminoglycoside-mediated damage is of urgent importance. Prostaglandin D2 synthase (*Ptgds*) is a glycoprotein that plays dual roles in lipid transport regulation and prostaglandin metabolism. However, the role of *Ptgds* in aminoglycoside-induced vestibular dysfunction remains unclear. This study aimed to explore the function of *Ptgds* in the utricle and HEI-OC1 cells. Neomycin injury induced high levels of *Ptgds* expression in utricle explants. Moreover, *Ptgds* knockdown protected against neomycin injury by enhancing cellular proliferation and viability while suppressing reactive oxygen species production, inflammation, and apoptosis. These findings suggest that *Ptgds* may serve as a novel therapeutic target for treating vestibular dysfunction caused by aminoglycoside-induced damage.

## Introduction

In humans, the vestibule plays a critical role in detecting head movements and facilitating the corresponding adjustment of trunk posture to maintain balance [1]. Vestibular hair cells are located in five key regions: the macula of the utricle, the macula of the saccule, and the ampullary crests of the three semicircular canals. Aminoglycosides, a class of antibiotics widely used in clinical practice, are associated with dose-dependent ototoxicity, which manifests as hair cell injury or death [2]. Damage to vestibular hair cells impairs vestibular function, leading to symptoms such as ataxia, impaired postural stability, dizziness, and vomiting [3]. Moderate-to-severe vertigo has been reported in up to 30% of affected patients [4]. Cochlear hair cells, once damaged or dead, are incapable of regeneration. While vestibular hair cells exhibit limited regenerative potential, restoring vestibular function after damage remains

**Data availability statement:** All relevant data are within the manuscript and its Supporting Information files.

**Funding:** This study was supported by National Natural Science Foundation of China (82000982, 82192865, 81970875). Guangdong Provincial Department of Science and Technology (A2022078). The funders had no role in study design, data collection and analysis, decision to publish, or preparation of the manuscript.

**Competing interests:** The authors have declared that no competing interests exist.

a significant challenge [5]. Currently, there are no effective treatments for aminoglycoside-induced vestibular dysfunction. Thus, investigating novel mechanisms and therapeutic targets involved in neomycin-associated vestibular hair cell damage holds great promise for advancing the clinical prevention and treatment of aminoglycoside-induced vestibular hair cell injury.

Prostaglandin D synthase (*Ptgds*) is a lipid transport glycoprotein primarily expressed in the heart, retina, and central nervous system. It catalyzes the production of prostaglandin D2 (PGD2) and facilitates the transport of lipophilic substances [6–9]. After synthesis, *Ptgds* undergoes modifications, enabling its secretion into the extracellular environment where it remains soluble in biofluids. Elevated levels of *Ptgds* expression have been reported in diffuse large B-cell lymphoma and other diseases [10]. Based on this, *Ptgds* was hypothesized to play a role in the pathogenesis of neomycin-induced vestibular hair cell injury in the inner ear.

Aminoglycosides are widely used to treat Gram-negative bacterial infections, including tuberculosis, respiratory infections, sepsis, and urinary tract infections. However, their clinical application is constrained by dose-dependent ototoxicity and vestibular injury [2]. Compared to cisplatin, aminoglycosides pose a higher risk of vestibular toxicity [11]. While vestibular hair cells possess some regenerative capacity, this ability is inherently limited. Regenerated vestibular hair cells often form immature hair bundles that fail to fully recover their functional properties after damage, contributing to persistent and irreversible vestibular dysfunction [12–14].

This study aimed to clarify the expression and functional role of *Ptgds* in neomycin-induced vestibular injury and to elucidate the underlying mechanisms. Our analyses provide novel evidence that *Ptgds* expression is upregulated in neomycin-damaged hair cells, potentially contributing to vertigo and other symptoms of vestibular dysfunction. Furthermore, *Ptgds* knockdown in utricle explants and HEI-OC1 cells was found to confer protection against neomycin-mediated injury. This protective effect is associated with a reduction in reactive oxygen species (ROS) production, suppression of inflammation, and inhibition of hair cell apoptosis. Collectively, these findings offer a new foundation for future research aimed at developing therapeutic strategies to mitigate aminoglycoside-induced vestibular dysfunction.

## Materials and methods

### Experimental mice and HEI-OC1 cells

Neonatal (postnatal day (p2)) and adult (p30) C57BL/6J mice were obtained from the Guangdong Medical Laboratory Animal Center. Both sexes of C57BL/6 J mice were used in this study at an equal ratio. The Institutional Review Board of Shenzhen University approved all animal studies. HEI-OC1 cells are an immortalized mouse cochlear cell line widely utilized in research on ototoxicity and auditory-related mechanisms.

### Utricle explant cultures and treatment

Mice were euthanized, decapitated, and their heads were placed in 75% ethanol before quickly being transferred into ice-cold PBS. Utricles were then excised, separated from the temporal bone, and cultured in 4-well culture dishes on coverslips coated with Cell-Tak while bathing these samples in DMEM/F12 containing N2, B27, and ampicillin. A model of hair cell injury was established by treating these utricles with neomycin (3 mM) for 12 h.

### CCK8

The CCK-8 (Cell Counting Kit-8) assay is utilized to evaluate cell viability or apoptosis. Initially, $1.2 \times 10^4$ cells per well are plated in a 96-well plate and permitted to adhere for 12 hours.

Experimental cohorts are then established, encompassing a neomycin concentration gradient (0–50mM, with a minimum of three replicates per group), a control group, a neomycin-damaged group, and a neomycin-damaged + siRNA group. Post 48-hour transfection, neomycin is administered. After 24 hours of neomycin exposure, a 10:1 mixture of DMEM and CCK-8 solution is prepared, with 100μL added to each well, followed by incubation at 37°C in the absence of light for 2 hours. Finally, absorbance at 450nm is measured using a microplate reader, and cell viability (%) is calculated as (OD experimental well - OD blank well) / (OD control well - OD blank well) × 100%.

## PCR and RNA-sequencing

TRIzol (Invitrogen) was used to extract total RNA, after which a transcriptor first-strand cDNA synthesis kit (Roche) was used for cDNA preparation. RNA expression was analyzed with a quantitative real-time PCR system (FastStart Universal SYBR Green Master Kit, Roche), using GAPDH as an endogenous control. The primers used to conduct this study are presented in Table 1.

An Agilent 2100 analyzer (Agilent Technologies, CA, USA) was used to assess RNA integrity, with only those samples exhibiting RNA integrity values > 7.0 being selected for RNA-sequencing (RNA-Seq). The Illumina HiSeq X Ten platform was used for cDNA sequencing. The R DESeq package (2012) was used to identify differentially expressed genes (DEGs) with the following criteria: P = 0.01, fold-change > 2, and FDR < 0.01.

## Cellular imaging

After immunofluorescence staining, images were captured using a Zeiss LSM 800 confocal scanning microscope. Excitation light at two different wavelengths, 488 nm and 750 nm, was employed. The image resolution was set to 1024 × 1024 pixels, utilizing 20× and 63× magnifications, and the scanning speed was adjusted to 8 (with modifications made as needed based on image quality). The utricle was divided into the striolar region and the extrastriolar region. In each utricle, random 100 μm × 100 μm areas were selected from both the striolar and extrastriolar regions for imaging. The number of myosin7a-positive cells was manually counted using ImageJ software (NIH, USA). CellROX-stained images were acquired at a resolution of 1024 × 1024 pixels under 20× magnification using a REVOLVE FL microscope (Discover Echo, USA).

## Immunofluorescence

Cells or tissue samples were fixed using 4% paraformaldehyde, permeabilized using 1% Triton X-100, blocked for 1 h with QuickBlockTM buffer (Beyotime, P0260), and probed overnight

**Table 1. The primers used to conduct this study are presented for PCR.**

| Gene | Forward | Reverse |
|---|---|---|
| *Caspase3* | AATCATGCCATTTGCCCAGC | CTCAAGTGTGTAGGGGGAGG |
| *Bcl2* | GTCGCTACCGTCGTGACTTC | CAGACATGCACCTACCCAGC |
| *Apaf1* | AGTGGCAAGGACACAGATGG | GGCTTCCGCAGCTAACACA |
| *Ptgds* | GAAGGCGGCCTCAATCTCAC | CGTACTCGTCATAGTTGGCCTC |
| *Gapdh* | TGGCCTTCCGTGTTCCTAC | GAGTTGCTGTTGAAGTCGCA |
| *iNOS* | GCATGGAACAGTATAAGGCAAACA | GTTTCTGGTCGATGTCATGAGCAA |
| *COX2* | GCATGGAACAGTATAAGGCAAACA | GTTTCTGGTCGATGTCATGAGCAA |
| *IL6* | TCCAGTTGCCTTCTTGGGAC | GTACTCCAGAAGACCAGAGG |

with anti-myosin 7a (Abcam, ab150386; 1:200) at 4°C. After washing thrice with PBS, cells were probed for 1 h at room temperature using appropriate secondary antibodies with Donkey anti-Rabbit IgG (H + L) Highly (invitrogen, A-31572), washed three additional times, and Fluoromount-G was then applied, followed by imaging with a confocal laser scanning microscope (Zeiss LSM 800, Germany).

### SiRNA transfection

Utricle explants and HEI-OC1 cells (1x10³/well) were added to 6-well plates, after which Lipofectamine RNAiMAX (13778075; Invitrogen) was used to transfect these samples with *Ptgds*-siRNA (50 nmol/L) followed by incubation for 48 h. Then, qPCR was used to assess transfection efficiency. The utilized siRNA sequences were as follows:*Ptgds*-siRNA,sense:5′-CAGUGUGAGACCAAGAUCAUG-3′,antisense: 5′- UGAUCUUGGUCUCACACUGGU-3′.

### Statistical analyses

Data were quantified as the mean ± standard error of the mean (SEM). Statistical analyses were conducted using Microsoft Excel and GraphPad Prism 9 software, with ImageJ utilized for cell counting within the immunofluorescence spectrum. Comparisons between two groups were performed using a two-tailed unpaired Student's t-test, whereas one-way ANOVA followed by Dunnett's multiple comparison test was employed for comparisons among two or more groups. A p-value of 0.05 was considered statistically significant.

## Results

### The effects of *Ptgds* expression

Initially, an in vitro model of neonatal rodent utricle culture was established, treating cells for 12 h with neomycin (3 mM) to induce neomycin-associated injury. RNA was then extracted from these injured utricle tissue samples for sequencing. Analyses of the identified gene expression patterns revealed 782 significant DEGs (Fig 1A), of which 534 and 248 were significantly upregulated and downregulated in neomycin-damaged utricle tissue samples relative to non-damaged samples. The *Ptgds* gene was found to be significantly upregulated in these tissue samples in response to neomycin-associated injury. We validated the RNA results through qRT-PCR experiments and found that *Ptgds* expression is upregulated in the utricle following neomycin-induced injury (S1 Fig).

HEI-OC1 cells were subsequently subjected to treatment with a range of neomycin concentrations (10, 20, 30, 40, or 50 mM), revealing that a 30 mM concentration resulted in a survival rate of ~50% such that this dose was selected for future experimental use when treating HEI-OC1 cells (Fig 1B). When qPCR was conducted to assess *Ptgds* gene expression in these samples, a greater than 20-fold increase in relative *Ptgds* levels was detected in response to treatment with a 30 mM neomycin dose (Fig 1C), indicating that *Ptgds* is significantly upregulated in vestibular hair cells and HEI-OC1 cells in response to neomycin exposure. Compared to the negative control group, the expression of Ptgds in HEI-OC1 cells was significantly suppressed by approximately 70% after 48 hours of siRNA transfection.

### *Ptgds* knockdown protects against neomycin-induced vestibular hair cell injury

To begin exploring the role that *Ptgds* plays in the context of neomycin-induced vestibular hair cell injury, a *Ptgds*-specific siRNA construct was generated and knockdown efficiency was

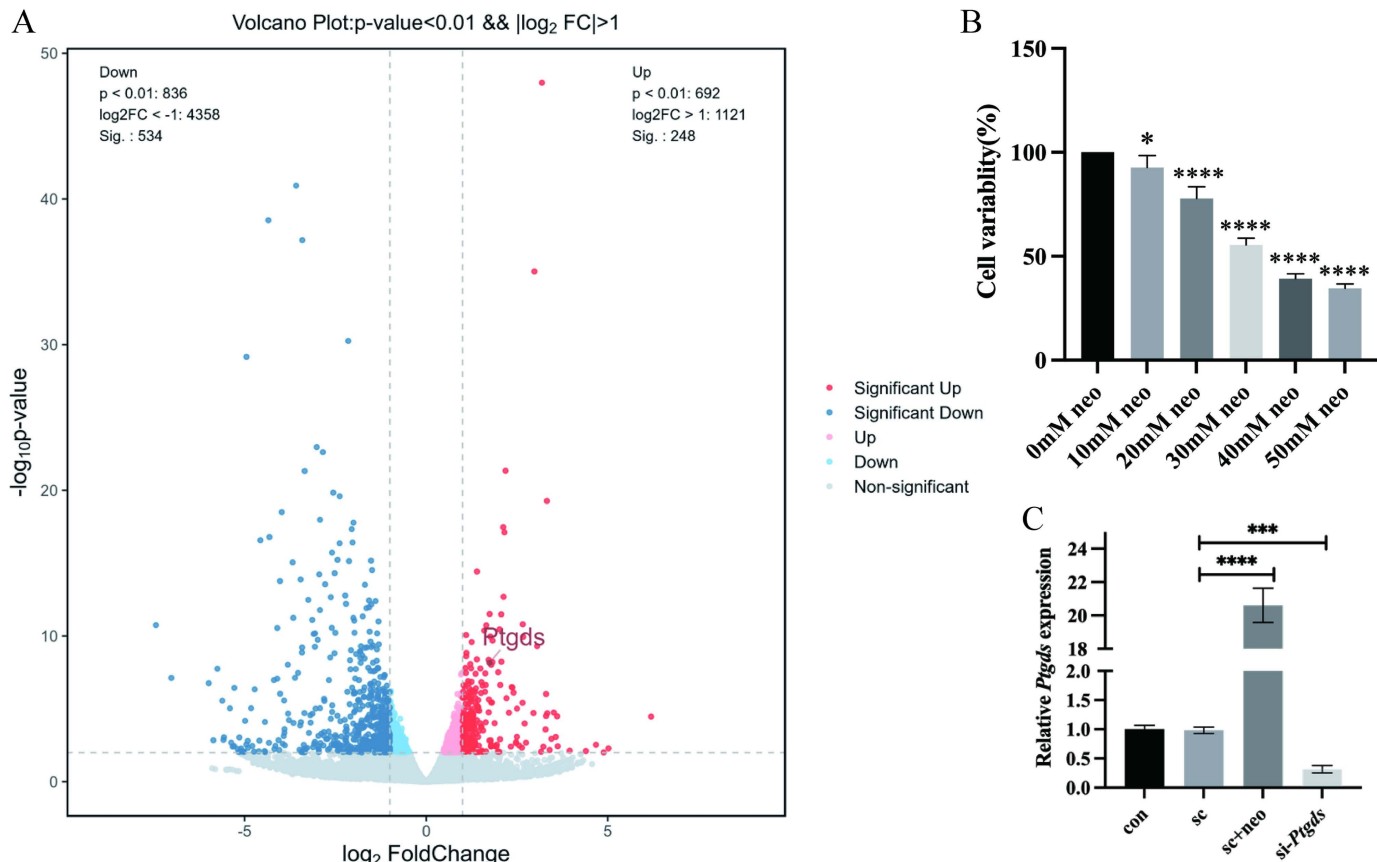

**Fig 1. Effects of *Ptgds* expression in neomycin-induced vestibular hair cells and HEI-OC1 cells.** Volcano plot representing gene-expression differ- ences neomycin group between control group in Utricle explant (**A**). The CCK8 method was used to determine the damage effect of neomycin (10, 20, 30, 40, and 50 mM) on HEI-OC1 cell (**B**). The qRT-PCR verifies the si-*Ptgds* transfection efficiency and the expression of *Ptgds* after neomycin injury in HEI-OC1 cells. The experimental groups included a blank control (con), a negative control (sc), a neomycin-induced damage group (sc+neo), and a *Ptgds* knockdown group silenced by siRNA for 48 hours (si-*Ptgds*). (**C**). The data are presented as the mean±SEM values. *P < 0.05, **P < 0.01, ***P < 0.001, ****P < 0.0001. (n = 3).

evaluated at 48 h post-transfection before 12 h with neomycin (3 mM) via qPCR. This approach ultimately achieved approximately 70% *Ptgds* knockdown (Fig 1C). The viability of cells in the control, neomycin-treated, and *Ptgds* knockdown groups was then assessed, revealing a ~19% increase in the viability of neomycin-treated cells in which *Ptgds* had been knocked down as compared to those in which it had not (76.9% vs. 57.71%)(Fig 2D). This suggests that knocking down *Ptgds* can protect against neomycin-induced injury to HEI-OC1 cells.

Numbers of hair cells in utricle explants treated under the same conditions in vitro were next quantified at 10x magnification (Fig 2A). These analyses revealed that neomycin-induced damage resulted in sparser hair cells, with an increase in overall hair cell density in explants in which *Ptgds* had been knocked down relative to the neomycin injury group. These explant were also assessed at higher magnification (63x), well-stained tissue areas from each group were selected, and the numbers of myosin 7a + cells in a 100 x 100 um area from the microstripe and non-microstripe area of each sample were counted. In the striolar region, few differences in hair cell numbers were observed between the control and siRNA control groups, whereas a significant drop in hair cell numbers was evident in the neomycin injury group, and this number rose significantly in the *Ptgds*-knockdown group as compared to the neomycin injury group (Fig 2B). In the extrastriolar region this same trend was observed (Fig 2C).

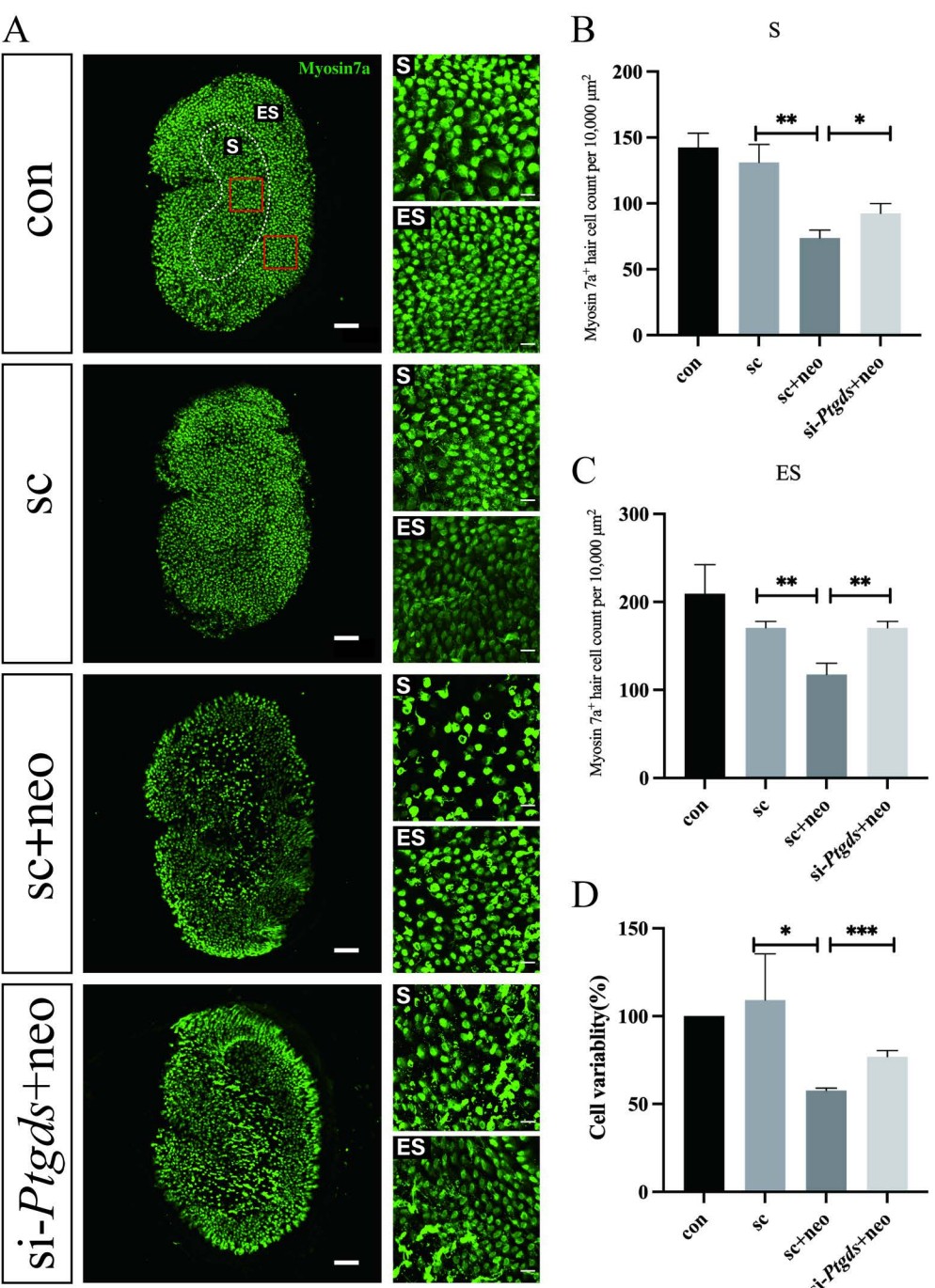

**Fig 2. Effects of knockdown *Ptgds* in neomycin-induced vestibular hair cells and HEI-OC1 cells.** Left, Immunostaining with myosin 7a in middle turns in Utricle explants. Scale bars = 100 μm (A). Right, magnified regions from left red boxes. Scale bars = 10 μm (A). S and ES indicate the striolar region and extrastriolar region.Graphical representation of the number of Myosin 7a + positive HCs per each treatment in striolar region (B). Graphical representation of the number of Myosin 7a + positive HCs per each treatment in extrastriolar region (C). The CCK8 method was used to determine the protective effect of si-*Ptgds* on 30mM neomycin-induced HEI-OC1 cell injury (D). The data are presented as the mean±SEM values. * P < 0.05, **P < 0.01, ***P < 0.001, ****P < 0.0001. (n = 3).

Immunostaining similarly confirmed the significant loss of hair cells in the striolar and extrastriolar regions of utricle explants that had been treated with 3 mM neomycin for 12 h. The knockdown of *Ptgds* expression significantly increased hair cell numbers in both of these regions following neomycin injury (Fig 2A–C). These data thus suggest that the silencing of *Ptgds* can protect against damage to neomycin-treated vestibular hair cells and HEI-OC1 cells.

### *Ptgds* knockdown suppresses neomycin-induced vestibular hair cell apoptotic death

TUNEL staining was further employed to explore the mechanisms through which the silencing of *Ptgds* protects against neomycin-induced injury. Utricle explants were initially cultured for 12 h in a tissue culture incubator (37°C, 5% CO2), followed by siRNA-mediated *Ptgds* knockdown. At 48 h post-transfection, samples were treated for 12 h with neomycin (3 mM). When examined via microscopy (63x), well-stained tissue areas from each group were selected, and the numbers of TUNEL+/myosin 7a+ cells in a 100 x 100 um area from each sample were counted. This approach revealed relatively few TUNEL+ hair cells in the control or siRNA control groups, whereas high numbers of these cells were evident in the neomycin injury group. *Ptgds* knockdown was associated with a significant reduction in the frequency of TUNEL+ myosin 7a-labeled hair cells relative to the neomycin injury group (Fig 3A, B). These data indicate that knocking down *Ptgds* can suppress neomycin-induced apoptotic death in vestibular cells, thereby protecting against injury.

Next, qPCR was used to evaluate changes in apoptosis-related gene expression in HEI-OC1 cells. Significant increases in levels of the pro-apoptotic mediators *Apaf-1* and *Caspase-3*

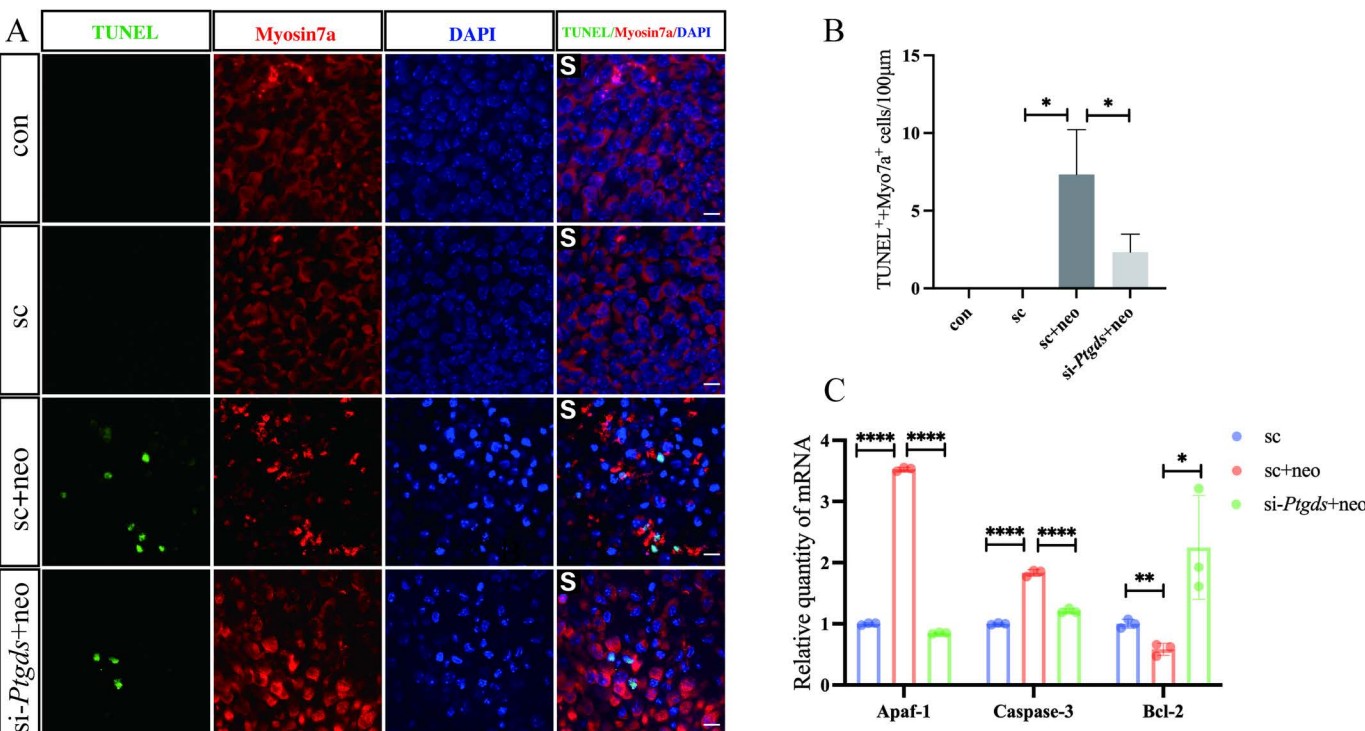

**Fig 3. Knockdown *Ptgds* reduces apoptosis of Vestibular hair cells and HEI-OC1 cells after neomycin treatment.** Immunostaining with TUNEL and myosin 7a labeling in the striolar region in Utricle explants. Scale bars = 10 μm (A). Graphical representation of the number of TUNEL highlighted HCs per each treatment in striolar region (B). The differentially expressed genes were verified in HEI- OC1 cells by qRT-PCR (C). The data are presented as the mean±SEM values. * P < 0.05, **P < 0.01, ***P < 0.001, ****P < 0.0001. (n = 3).

were observed in neomycin-damaged cells relative to control cells, with the concomitant downregulation of anti-apoptotic factors. *Ptgds* knockdown significantly reduced and increased the levels of respective pro- and anti-apoptotic mediators relative to those observed in neomycin-injured cells. These data thus demonstrate that *Ptgds* silencing can protect HEI-OC1 cells against neomycin-induced apoptotic death and associated damage.

## *Ptgds* knockdown abrogates neomycin-induced ROS production in HEI-OC1 cells

As ROS accumulation is closely tied to the neomycin-induced apoptotic death of hair cells, the CellROX green probe was utilized to evaluate the production of mitochondrial ROS within HEI-OC1 cells. Neomycin-induced damage was associated with elevated ROS levels as evidenced by an increase in green fluorescent signal intensity relative to the control and *Ptgds*-knockdown groups, with the silencing of *Ptgds* having significantly suppressed ROS treatment in response to neomycin treatment (Fig 4A, B). These results highlight the fact that silencing *Ptgds* can significantly mitigate oxidative stress resulting from neomycin treatment in HEI-OC1 cells.

## *Ptgds* knockdown suppresses the neomycin-induced production of inflammatory mediators by HEI-OC1 cells

To further examine the relationship between neomycin-induced damage and inflammatory activity in hair cells, qPCR was next conducted. This approach revealed significant increases

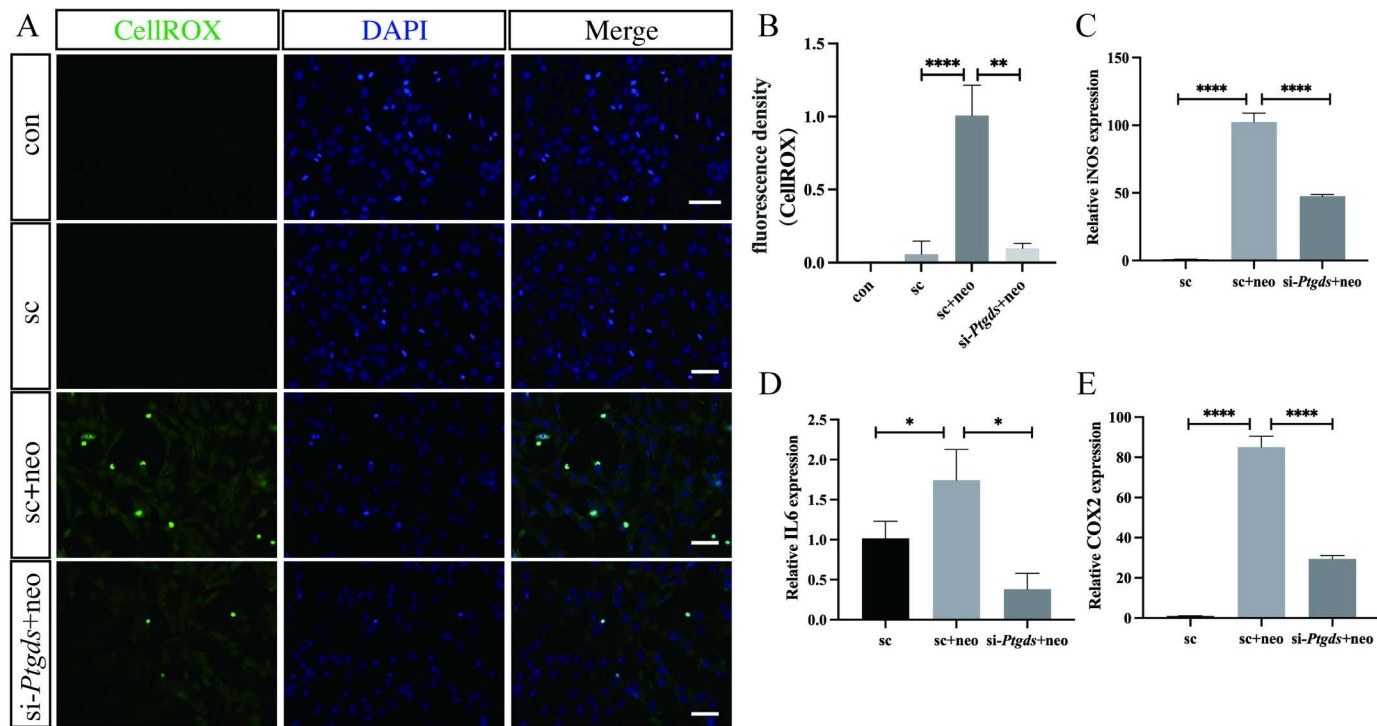

**Fig 4. Knockdown *Ptgds* reduces ROS and inflammatory factor Expression of HEI-OC1 cells after neomycin treatment. CellROXTM green staining of HEI-OC1 cells. Scale bars = 20 μm.** (A). Graphical representation of the density of CellROX highlighted HEI-OC1 cells per each treatment (B). The differentially expressed genes were verified in HEI- OC1 cells by qRT-PCR(C-E). The data are presented as the mean±SEM values. *P < 0.05, **P < 0.01, ***P < 0.001, ****P < 0.0001. (n = 3).

in *IL6*, *iNOS*, and *COX2* mRNA levels in response to neomycin treatment, whereas the expression of these genes was significantly reduced by *Ptgds* knockdown (Fig 4C–E). This suggests that the loss of *Ptgds* may protect against neomycin-induced inflammatory activity within hair cells.

## Discussion

Aminoglycosides (AGs) are a class of antibiotics widely used in clinical practice, known not only for their ototoxicity but also for their vestibulotoxicity. During clinical administration, AGs frequently cause damage to vestibular hair cells, which can lead to vestibular dysfunction. This dysfunction manifests as clinical symptoms such as ataxia, postural instability, dizziness, and vomiting. Neomycin, one of the commonly used aminoglycosides, is typically administered topically or orally in clinical settings. These routes of administration result in relatively low plasma concentrations of neomycin, thereby minimizing systemic exposure. However, despite the lower systemic absorption, neomycin is still associated with significant nephrotoxicity and ototoxicity. Due to these adverse effects, intravenous administration of neomycin is rarely employed in clinical practice [2].

While neomycin demonstrates favorable therapeutic efficacy, its clinical application is severely limited by its unavoidable side effects, particularly its toxicity to the auditory and vestibular systems. In this study, the concentration of neomycin used exceeds the typical plasma concentrations observed during clinical use. However, it is consistent with concentrations commonly employed in in vitro studies designed to model ototoxicity. This approach allows for a more accurate investigation of the mechanisms underlying aminoglycoside-induced vestibular damage, providing valuable insights that may contribute to the development of strategies to mitigate these adverse effects in clinical practice.

This study is the first to demonstrate the significant upregulation of *Ptgds* in utricle explants and HEI-OC1 cells following neomycin-induced injury. The knockdown of *Ptgds* expression may protect vestibular hair cells in the inner ear and attenuate neomycin-induced vestibular toxicity. In vitro experiments conducted in this study revealed that *Ptgds* silencing effectively mitigated neomycin-induced ROS production and apoptotic cell death in hair cells. This protective effect was also associated with the suppression of inflammatory activity, as evidenced by the reduced upregulation of COX2, IL6, and iNOS in response to neomycin when *Ptgds* expression was silenced. Notably, the loss of *Ptgds* may also confer protective benefits to tumor cells by limiting inflammation and suppressing deleterious immune responses.

*Ptgds* is a glycoprotein that serves dual roles in catalyzing *PGD2* synthesis and transporting lipophilic substances [6]. Increasing evidence suggests that *Ptgds* is upregulated in various cancers, including melanoma, lung cancer, gastric cancer, and diffuse large B-cell lymphoma, underscoring its potential as a therapeutic target in oncology [10,15–18]. However, the role of *Ptgds* in vestibular toxicity and ototoxicity remains poorly understood. In this study, *Ptgds* knockdown was sufficient to protect utricle explants and HEI-OC1 cells against neomycin-induced damage (Fig 2A, B). These findings highlight the potential utility of *Ptgds* as a genetic target for preventing aminoglycoside-induced vestibular toxicity.

Aminoglycosides induce toxic effects on vestibular hair cells primarily through necrotic and apoptotic pathways [19,20]. In this study, TUNEL staining revealed a significant increase in TUNEL/myosin 7a double-positive cells in utricle explants treated with neomycin in vitro, indicating that neomycin promotes apoptotic death in vestibular hair cells. Strikingly, *Ptgds* silencing largely abrogated this TUNEL staining following neomycin exposure (Fig 3A). In addition, qPCR analysis showed that neomycin treatment upregulated pro-apoptotic mediators such as Apaf-1 and caspase-3 while downregulating the anti-apoptotic mediator Bcl-2, consistent with the induction of apoptosis in these cells. Knockdown of *Ptgds* reversed these

changes in apoptotic mediator expression, suggesting that *Ptgds* silencing significantly reduces neomycin-induced hair cell apoptosis in both the striolar and extrastriolar regions.

The production of elevated ROS levels is another key mechanism underlying aminoglycoside-induced vestibular toxicity. Previous studies have shown that aminoglycoside-induced ROS production in hair cells contributes to cell death through the activation of the JNK pathway and/or increased mitochondrial permeability, ultimately triggering apoptosis [21–23]. Consistent with these reports, neomycin treatment resulted in significant ROS production in HEI-OC1 cells, as detected via CellROX staining, while *Ptgds* silencing significantly reduced these levels (Fig 3B). Collectively, these findings demonstrate that *Ptgds* knockdown protects hair cells from neomycin-induced injury by reducing ROS production and inhibiting apoptotic cell death.

Cisplatin-induced inner ear injury has been shown to provoke immune responses, including the release of cytokines that damage hair cells and exacerbate vestibular dysfunction [1,24,25]. In this study, we hypothesized that neomycin induces a similar inflammatory response in inner ear hair cells. Supporting this hypothesis, neomycin-treated HEI-OC1 cells exhibited increased mRNA expression of IL6, COX2, and iNOS. However, *Ptgds* silencing significantly suppressed the upregulation of these inflammatory mediators, suggesting that *Ptgds* silencing protects against neomycin-induced injury by mitigating inflammatory responses in hair cells.

While the precise cellular targets of *Ptgds* and its mechanisms of action in protecting against drug-induced toxicity remain unclear, this study provides the first evidence for the involvement of *Ptgds* in neomycin-induced vestibular dysfunction. Inhibiting *Ptgds* expression protects against neomycin-induced vestibular injury by suppressing ROS production, inflammation, and apoptotic cell death. These findings have broad implications, highlighting the potential of *Ptgds* as a therapeutic target for addressing vestibular dysfunction.

## Supporting Information

**S1 Fig. Effects of Ptgds expression in neomycin-induced vestibular hair cells and HEI-OC1 cells.** Validation of RNA results by qRT-PCR showing upregulation of Ptgds expression in the utricle following 3mM neomycin-induced injury.
(TIFF)

**S1 Table. The results of the one-way ANOVA for each figure panel are summarized as follows.** In Fig 1B, a highly significant treatment effect was observed (F = 265.6, P < 0.0001), indicating substantial differences between the treatment groups. Fig 1C also showed a significant effect (F = 1115, P < 0.0001), with distinct group differences. In Fig 2B, a large effect was found (F = 30.95, P < 0.0001), while Fig 2C and Fig 2D demonstrated moderate effects (F = 12.63, P = 0.0021 and F = 9.216, P = 0.0057, respectively). Fig 3B showed significant differences (F = 14.84, P = 0.0012), and Fig 3C-apaf1 had an exceptionally strong effect (F = 14260, P < 0.0001). For Fig 3C-cas3, the treatment effect was highly significant (F = 371.8, P < 0.0001), while Fig 3C-bcl2 revealed a moderate effect (F = 9.188, P = 0.0149). In Fig 4B, a strong treatment effect was observed (F = 52.65, P < 0.0001), and Fig 4C showed a very large treatment effect (F = 531.0, P < 0.0001). Finally, Fig 4D and Fig 4E exhibited significant treatment effects with moderate to large F values (F = 13.47, P = 0.0097 and F = 521.6, P < 0.0001, respectively). All the results demonstrate robust treatment effects with varying degrees of significance across the panels.
(PDF)

## Author contributions

**Formal analysis:** Zhimin Zhao.

**Funding acquisition:** Guohui Nie.

**Investigation:** Chen Chen, Zhimin Zhao.

**Methodology:** Jinghong Han.

**Project administration:** Jinghong Han.

**Software:** Yue Zhang.

**Validation:** Yue Zhang.

**Writing – original draft:** Chen Chen.

**Writing – review & editing:** Guohui Nie.

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
