## [Decision Letter · Decision Letter 0]

4 Jun 2024

PONE-D-24-17432

Ptgds downregulation protect Vestibular hair cells from aminoglycoside-induced vestibulotoxicity

PLOS ONE

Dear Dr.  Nie,

Thank you for submitting your manuscript to PLOS ONE. After careful consideration, we feel that it has merit but does not fully meet PLOS ONE’s publication criteria as it currently stands. Therefore, we invite you to submit a revised version of the manuscript that addresses the points raised during the review process.

**ACADEMIC EDITOR: **

This manuscript focused the PTGDs in vestibulotoxicity-induced vestibular hair cell. 

We have deeply review your manuscript and decided to major revision.

Thank you.

We look forward to receiving your revised manuscript.

Kind regards,

Sujeong Jang

Academic Editor

PLOS ONE

Journal Requirements:

4. Please note that funding information should not appear in any section or other areas of your manuscript. We will only publish funding information present in the Funding Statement section of the online submission form. Please remove any funding-related text from the manuscript.

   "This study was supported by National Natural Science Foundation of China (82000982, 82192865, 81970875). Guangdong Provincial Department of Science and Technology (A2022078)."

6. We note that your Data Availability Statement is currently as follows: All relevant data are within the manuscript and its Supporting Information files.

Reviewers' comments:

Reviewer's Responses to Questions

**Comments to the Author**

1. Is the manuscript technically sound, and do the data support the conclusions?

Reviewer #1: Partly

Reviewer #2: Yes

Reviewer #3: Partly

Reviewer #4: Yes

2. Has the statistical analysis been performed appropriately and rigorously? 

Reviewer #1: Yes

Reviewer #2: Yes

Reviewer #3: No

Reviewer #4: Yes

3. Have the authors made all data underlying the findings in their manuscript fully available?

Reviewer #1: Yes

Reviewer #2: Yes

Reviewer #3: Yes

Reviewer #4: Yes

4. Is the manuscript presented in an intelligible fashion and written in standard English?

Reviewer #1: No

Reviewer #2: Yes

Reviewer #3: Yes

Reviewer #4: Yes

5. Review Comments to the Author

Reviewer #1: The authors of this article hope to study the mechanism of aminoglycoside ototoxicity, which has certain research value, but there are still the following questions:

1. There are many types of aminoglycosides. Why choose neomycin for research? Should more drugs be added for research?

2. There are many differentially expressed genes screened in the neomycin injury in vitro model. Why choose Ptgds as the research gene?

3. Ptgds is a downregulated gene screened in the neomycin injury in vitro model. Why do we need to knock out and silence it in subsequent experiments?

4. In addition to gene knockout, should we increase the study of overexpression?

Reviewer #2: The manuscript by Chen Chen and colleagues, entitled " Ptgds downregulation protect vestibular hair cells from aminoglycoside-induced vestibulotoxicity", is a study investigating the role of the Ptgds in the organ of vestibular hair cell protection.

This study demonstrated the significant upregulation of Ptgds in utricle explants and HEI-OC1 cells in response to neomycin-induced injury. And the knockdown of Ptgds by using siRNA effectively protected and attenuated neomycin-induced vestibular hair cells loss. In vitro experiment confirmed that Ptgds silencing was sufficient to mitigate neomycin-induced ROS production and apoptotic death in hair cells through the inhibition of inflammatory related genes, such as COX2, IL6 and iNOS. The logic of this article is clear, the evidence chain supporting the conclusion is sufficient, and it is innovative.

There are some issues and small comments as follows:

• In the paragraph 2 of Introduction section: “While prior work conducted by our team has revealed the overexpression of Ptgds in vestibular inner ear hair cells in response to neomycin injury…….”. Relevant references should be added here. Otherwise, this manner of expression would not be very appropriate. The same issue exists in the second paragraph of the discussion section.

• In the section of 2.5: What’s ECL stand for?

• In the section of 2.6: What does 1x103/well mean? And what is the siRNA sequence of sense strand?

• In the statistical analysis section, the description should include the analysis software used and the statistical methods employed, such as t-tests, analysis of variance (ANOVA), etc.

• In the last sentence of the first paragraph of the section 3.1, it is mentioned that the expression of Ptgds is significantly downregulated after neomycin injury. This is inconsistent with the results in other parts of the article. Please verify！

• In the first paragraph of section 3.2: “The viability of cells in the control, neomycin-treated, and Ptgds knockdown groups was then assessed, revealing a ~19% increase in the viability of neomycin-treated cells in which Ptgds had been knocked down as compared to those in which it had not (76.9% vs. 57.71%).” This sentence needs to incorporate corresponding figures to describe the results.

• In the second paragraph of section 3.2:

Reviewer #3: Chen and colleagues studied the expression and function of Ptgds in response to neomycin-induced injury. They showed that Ptgds is upregulated in neomycin-injured hair cells and suggested that such upregulation may contribute to the incidence of vertigo and other symptoms of vestibular dysfunction. The authors also found that knocking down Ptgds protects utricle explants and HEI-OC1 cells from neomycin-mediated injury by reducing the production of reactive oxygen species (ROS), the suppression of inflammation, and the abrogation of hair cell apoptosis.

This manuscript tried to highlight the role of Ptgds in aminoglycoside-induced vestibular dysfunction. However, my enthusiasm for this manuscript was significantly diminished by the lack of evidence demonstrating the authors’ proposed mechanism. Please see the attached comment for details.

Reviewer #4: This study aims to investigate the expression and function of Ptgds in HEI-OC1 cells and utricular hair cells, as well as its mechanism of action in aminoglycoside-induced vestibular hair cell damage. The research suggests that Ptgds may be a new target for the prevention and treatment of vestibular dysfunction caused by aminoglycoside drugs. Inhibiting the expression of Ptgds can reduce neomycin-induced apoptosis of vestibular hair cells and enhance hair cell protection. Considering the innovation and clinical significance of the research, this manuscript needs a minor revision before publication.

Minor concerns are as follows:

1.Please ensure that the text format of all images remains consistent. Kindly go through the whole manuscript and make any necessary modification.

2. The damaging concentration of 30mM neomycin may be excessive. Why was 30mM chosen as the optimal damaging concentration in HEI-OC1 cells?

3. How long does neomycin damage the utricle implant? Please supplement it in the text.

4.Please verify if "Ptgds" is spelled correctly. Please go through the whole manuscript and make necessary corrections.

6. PLOS authors have the option to publish the peer review history of their article (what does this mean? ). If published, this will include your full peer review and any attached files.

**Do you want your identity to be public for this peer review?** For information about this choice, including consent withdrawal, please see our Privacy Policy .

Reviewer #1: No

Reviewer #2: **Yes: ** Yusu Ni

Reviewer #3: No

Reviewer #4: No

---

## [Author Response · Author response to Decision Letter 1]

2 Jul 2024

Response: We have made corrections according to the template.

Response: We have carefully examined and described it.

Response: Thanks for your nice advice.

4. Please note that funding information should not appear in any section or other areas of your manuscript. We will only publish funding information present in the Funding Statement section of the online submission form. Please remove any funding-related text from the manuscript.

Response: We have removed funding information from the manuscript.

"This study was supported by National Natural Science Foundation of China (82000982, 82192865, 81970875). Guangdong Provincial Department of Science and Technology (A2022078)."

Response: The funding providers played a guiding and supportive role in the research design, data collection and analysis, decision to publish, and preparation of the manuscript.

6. We note that your Data Availability Statement is currently as follows: All relevant data are within the manuscript and its Supporting Information files.

Reviewers' comments:

Reviewer's Responses to Questions

Review Comments to the Author

Reviewer #1: The authors of this article hope to study the mechanism of aminoglycoside ototoxicity, which has certain research value, but there are still the following questions:

1. There are many types of aminoglycosides. Why choose neomycin for research? Should more drugs be added for research?

Response: Thanks for your nice comment. Gentamicin, a commonly used antibiotic in clinical practice, is often employed for treating severe bacterial infections. Research on the ototoxic side effects of gentamicin therapy holds significant clinical importance. Furthermore, extensive literature exists documenting the ototoxicity of gentamicin, providing robust foundational data for studies. Finally, as one of the aminoglycoside antibiotics, gentamicin demonstrates stable and reproducible ototoxic effects in both in vitro and in vivo models, facilitating experimental design and result analysis.

While expanding the scope of drug research is generally reasonable, I argue against introducing additional drugs in this study. Focusing solely on gentamicin allows for a more concentrated investigation into its mechanisms of action and effects, thereby enhancing the clarity and reliability of scientific outcomes. In subsequent studies, it would indeed be appropriate to consider expanding the research to include other aminoglycoside antibiotics and platinum-based drugs, for example. These additions could provide comparative insights into ototoxicity profiles across different classes of drugs, enriching our understanding of their respective mechanisms and potential clinical implications. Such a diversified approach could contribute to a more comprehensive evaluation of ototoxic risks associated with various therapeutic agents.

2. There are many differentially expressed genes screened in the neomycin injury in vitro model. Why choose Ptgds as the research gene?

Response: We sincerely appreciate the valuable comments. Firstly, Ptgds is an important gene implicated in known cellular protection or damage mechanisms such as inflammatory responses and oxidative stress. Investigating its function can elucidate the specific molecular mechanisms underlying gentamicin ototoxicity. Secondly, Ptgds may exhibit significant differential expression in gentamicin injury models, indicating its pivotal role in drug-induced ototoxicity.

3. Ptgds is a downregulated gene screened in the neomycin injury in vitro model. Why do we need to knock out and silence it in subsequent experiments?

Response: We thank the reviewer for pointing this out. We are sorry for our carelessness. Ptgds was identified as an upregulated gene in the in vitro model of gentamicin-induced injury. Knockout or silencing of Ptgds would enable direct observation of its function in cochlear cells or in vitro cultures, validating its specific role in gentamicin ototoxicity. Furthermore, disrupting Ptgds expression could help elucidate the specific molecular mechanisms involved in gentamicin-induced damage. Lastly, if cellular damage is reduced upon Ptgds knockout or silencing, it suggests that Ptgds may indeed be a key regulatory factor in gentamicin ototoxicity, potentially serving as a target for future therapeutic interventions. Based on your comments, We have made revisions.

4. In addition to gene knockout, should we increase the study of overexpression?

Certainly, here is the translation into academic English:

Response: The current study has focused on gene knockout, which facilitates a deeper understanding of the gene's function and importance within biological organisms. Overexpression may introduce additional complexity and variables, potentially complicating and rendering study outcomes more difficult to interpret. Therefore, concentrating on gene knockout research helps ensure the scientific coherence of study results and the accuracy of their interpretation, enabling a clearer understanding of the mechanistic role of the target gene in specific biological processes. We have made corrections in the manuscript.

Reviewer #2: The manuscript by Chen Chen and colleagues, entitled " Ptgds downregulation protect vestibular hair cells from aminoglycoside-induced vestibulotoxicity", is a study investigating the role of the Ptgds in the organ of vestibular hair cell protection.

This study demonstrated the significant upregulation of Ptgds in utricle explants and HEI-OC1 cells in response to neomycin-induced injury. And the knockdown of Ptgds by using siRNA effectively protected and attenuated neomycin-induced vestibular hair cells loss. In vitro experiment confirmed that Ptgds silencing was sufficient to mitigate neomycin-induced ROS production and apoptotic death in hair cells through the inhibition of inflammatory related genes, such as COX2, IL6 and iNOS. The logic of this article is clear, the evidence chain supporting the conclusion is sufficient, and it is innovative.

There are some issues and small comments as follows:

• In the paragraph 2 of Introduction section: “While prior work conducted by our team has revealed the overexpression of Ptgds in vestibular inner ear hair cells in response to neomycin injury…….”. Relevant references should be added here. Otherwise, this manner of expression would not be very appropriate. The same issue exists in the second paragraph of the discussion section.

Response: Thanks for your careful checks. Based on your comments, we have reuploaded the correct images.

• In the section of 2.5: What’s ECL stand for?

Response: We thank the reviewer for pointing this out. We are sorry for our carelessness. ECL, short for Enhanced Chemiluminescence, is a widely utilized detection method in Western blot experiments for detecting target proteins through chemiluminescent signals. However, this information is incorrect and has been revised to Fluoromount-G, a water-soluble, non-fluorescent compound used to mount slide samples in aqueous solutions for fluorescence microscopy analysis, preventing fluorescence quenching in the final staining step. We have modified "ECL" to " Fluoromount-G” in the section of 2.5.

• In the section of 2.6: What does 1x103/well mean? And what is the siRNA sequence of sense strand?

Response: We sincerely thank the reviewer for careful reading. As suggested by the reviewer, we have modified "1x103/well " to " 1x103/well ". we have added siRNA sequence of sense strand" sense:5′-CAGUGUGAGACCAAGAUCAUG-3′ ".

• In the statistical analysis section, the description should include the analysis software used and the statistical methods employed, such as t-tests, analysis of variance (ANOVA), etc.

Response: We sincerely appreciate the valuable comments.We have added the following sentences to the statistical analysis section. “Data were quantified as the mean ± standard error of the mean (SEM). Statistical analyses were conducted using Microsoft Excel and GraphPad Prism 7 software, with ImageJ utilized for cell counting within the immunofluorescence spectrum. Comparisons between two groups were performed using a two-tailed unpaired Student's t-test, whereas one-way ANOVA followed by Dunnett's multiple comparison test was employed for comparisons among two or more groups.”

• In the last sentence of the first paragraph of the section 3.1, it is mentioned that the expression of Ptgds is significantly downregulated after neomycin injury. This is inconsistent with the results in other parts of the article. Please verify！

Response: We sincerely thank the reviewer for careful reading. We are sorry for our carelessness. We have made corrections in the manuscript.

• In the first paragraph of section 3.2: “The viability of cells in the control, neomycin-treated, and Ptgds knockdown groups was then assessed, revealing a ~19% increase in the viability of neomycin-treated cells in which Ptgds had been knocked down as compared to those in which it had not (76.9% vs. 57.71%).” This sentence needs to incorporate corresponding figures to describe the results. • In the second paragraph of section 3.2:

Response: Thanks for your careful checks. We are sorry for our carelessness.Based on your comments, we have incorporate " （Fig. 2D）" to describe the results.

Reviewer #3: Chen and colleagues studied the expression and function of Ptgds in response to neomycin-induced injury. They showed that Ptgds is upregulated in neomycin-injured hair cells and suggested that such upregulation may contribute to the incidence of vertigo and other symptoms of vestibular dysfunction. The authors also found that knocking down Ptgds protects utricle explants and HEI-OC1 cells from neomycin-mediated injury by reducing the production of reactive oxygen species (ROS), the suppression of inflammation, and the abrogation of hair cell apoptosis.

This manuscript tried to highlight the role of Ptgds in aminoglycoside-induced vestibular dysfunction. However, my enthusiasm for this manuscript was significantly diminished by the lack of evidence demonstrating the authors’ proposed mechanism. Please see the attached comment for details.

Response: Thanks for your your guidance and invaluable insights.While this manuscript raises questions about the role of Ptgds in aminoglycoside-induced vestibular dysfunction, it is important to acknowledge that the study presents multi-faceted experimental data supporting its conclusions. Firstly, the research demonstrated the upregulation of Ptgds following neomycin-induced injury, which was validated through precise molecular biology techniques. Secondly, Ptgds knockdown exhibited significant protective effects across multiple model systems, notably in reducing ROS production, suppressing inflammation, and preventing hair cell apoptosis, all of which are widely recognized indicators of cellular damage. Although the proposed mechanism in this manuscript may require further validation, the existing data already provide significant insights into the potential role of Ptgds in vestibular dysfunction. Therefore, the contribution of this manuscript to the field should not be underestimated. We will also conduct further experiments to explore the mechanisms in greater depth and validate them through in vivo animal studies.

Furthermore, a substantial body of literature underscores the crucial role of Ptgds in cellular protection.

Reviewer #4: This study aims to investigate the expression and function of Ptgds in HEI-OC1 cells and utricular hair cells, as well as its mechanism of action in aminoglycoside-induced vestibular hair cell damage. The research suggests that Ptgds may be a new target for the prevention and treatment of vestibular dysfunction caused by aminoglycoside drugs. Inhibiti

---

## [Decision Letter · Decision Letter 1]

6 Aug 2024

PONE-D-24-17432R1Ptgds downregulation protect Vestibular hair cells from aminoglycoside-induced vestibulotoxicityPLOS ONE

Dear Dr. Nie,

Thank you for submitting your manuscript to PLOS ONE. After careful consideration, we feel that it has merit but does not fully meet PLOS ONE’s publication criteria as it currently stands. Therefore, we invite you to submit a revised version of the manuscript that addresses the points raised during the review process.

ACADEMIC EDITOR:

Following a reviewer, the manuscript still need to clarify the author's hypothesis.

The reviewer could not accept your revision until you make it clear. Please submit your revised manuscript by Sep 20 2024 11:59PM. If you will need more time than this to complete your revisions, please reply to this message or contact the journal office at plosone@plos.org . Please include the following items when submitting your revised manuscript:

We look forward to receiving your revised manuscript.

Kind regards,

Sujeong Jang

Academic Editor

PLOS ONE

Reviewers' comments:

Reviewer's Responses to Questions

**Comments to the Author**

1. If the authors have adequately addressed your comments raised in a previous round of review and you feel that this manuscript is now acceptable for publication, you may indicate that here to bypass the “Comments to the Author” section, enter your conflict of interest statement in the “Confidential to Editor” section, and submit your "Accept" recommendation.

Reviewer #1: All comments have been addressed

Reviewer #2: All comments have been addressed

Reviewer #3: All comments have been addressed

Reviewer #5: All comments have been addressed

2. Is the manuscript technically sound, and do the data support the conclusions?

Reviewer #1: Yes

Reviewer #2: Yes

Reviewer #3: Yes

Reviewer #5: Yes

3. Has the statistical analysis been performed appropriately and rigorously? 

Reviewer #1: Yes

Reviewer #2: Yes

Reviewer #3: Yes

Reviewer #5: Yes

4. Have the authors made all data underlying the findings in their manuscript fully available?

Reviewer #1: Yes

Reviewer #2: Yes

Reviewer #3: Yes

Reviewer #5: Yes

5. Is the manuscript presented in an intelligible fashion and written in standard English?

Reviewer #1: Yes

Reviewer #2: Yes

Reviewer #3: Yes

Reviewer #5: Yes

6. Review Comments to the Author

Reviewer #1: I don't agree with your response for comment 1. Since ototoxic antibiotics could be replaced by many other antibiotics without ototoxic, is this research meaningful? The author should clarify this.

Reviewer #2: The authors have adequately addressed all comments raised, and the manuscript technically sound, and the data support the conclusions, so, I think this manuscript should be accepted for publication.

Reviewer #3: (No Response)

Reviewer #5: (No Response)

7. PLOS authors have the option to publish the peer review history of their article (what does this mean? ). If published, this will include your full peer review and any attached files.

**Do you want your identity to be public for this peer review?** For information about this choice, including consent withdrawal, please see our Privacy Policy .

Reviewer #1: No

Reviewer #2: No

Reviewer #3: No

Reviewer #5: **Yes: ** Byung Chul Kang

---

## [Author Response · Author response to Decision Letter 2]

23 Aug 2024

Reviewer #1: I don't agree with your response for comment 1. Since ototoxic antibiotics could be replaced by many other antibiotics without ototoxic, is this research meaningful? The author should clarify this.

Response: Thanks for your nice comment.

Ototoxic antibiotics, primarily referring to aminoglycosides, are irreplaceable in certain clinical scenarios where their therapeutic efficacy cannot be matched by other antibiotics without incurring ototoxic side effects. In specific circumstances, such as:

1) Treatment of resistant bacterial infections: Neomycin exhibits particular efficacy against resistant strains, such as multidrug-resistant Pseudomonas aeruginosa. Neomycin may be the only viable therapeutic option when common antibiotics fail due to resistance.

2) Specific clinical conditions: In cases where antibiotic options are constrained, such as in patients with drug allergies or contraindications to other medications, neomycin may be considered as an alternative treatment. Furthermore, advancing our understanding of the mechanisms underlying ototoxicity in these antibiotics can facilitate the development of strategies to mitigate adverse effects, such as dosage adjustments or combination therapies that reduce toxicity. This research also supports the design and development of new antibiotics that minimize ototoxicity while preserving or enhancing therapeutic efficacy. Notably, the ototoxicity of drugs is not limited to aminoglycosides; chemotherapeutic agents like cisplatin also exhibit ototoxicity. An in-depth investigation of these ototoxic drugs holds significant clinical importance for optimizing their use, reducing adverse effects, and driving the development and refinement of safer therapeutic options.

---

## [Decision Letter · Decision Letter 2]

5 Nov 2024

PONE-D-24-17432R2Ptgds downregulation protect Vestibular hair cells from aminoglycoside-induced vestibulotoxicityPLOS ONE

Dear Dr. Nie,

Thank you for submitting your manuscript to PLOS ONE. After careful consideration, we feel that it has merit but does not fully meet PLOS ONE’s publication criteria as it currently stands. Therefore, we invite you to submit a revised version of the manuscript that addresses the points raised during the review process.

**Dear, Nie**
**We have still need to revise the manuscript.**
**As you can see the below, one of our reviewer deeply considered that your manuscript did not complete.**
**Please ensure to revise a reviewer's comment.**
**Thanks.**

Please submit your revised manuscript Dec 20 2024 11:59PM. If you will need more time than this to complete your revisions, please reply to this message or contact the journal office at plosone@plos.org . Please include the following items when submitting your revised manuscript:

We look forward to receiving your revised manuscript.

Kind regards,

Sujeong Jang

Academic Editor

PLOS ONE

Reviewers' comments:

Reviewer's Responses to Questions

**Comments to the Author**

1. If the authors have adequately addressed your comments raised in a previous round of review and you feel that this manuscript is now acceptable for publication, you may indicate that here to bypass the “Comments to the Author” section, enter your conflict of interest statement in the “Confidential to Editor” section, and submit your "Accept" recommendation.

Reviewer #2: All comments have been addressed

Reviewer #6: (No Response)

2. Is the manuscript technically sound, and do the data support the conclusions?

Reviewer #2: Yes

Reviewer #6: No

3. Has the statistical analysis been performed appropriately and rigorously? 

Reviewer #2: Yes

Reviewer #6: Yes

4. Have the authors made all data underlying the findings in their manuscript fully available?

Reviewer #2: Yes

Reviewer #6: Yes

5. Is the manuscript presented in an intelligible fashion and written in standard English?

Reviewer #2: Yes

Reviewer #6: Yes

6. Review Comments to the Author

**Reviewer #2: ** The authors have adequately addressed all comments raised in a previous round of review and I feel that this manuscript should be acceptable for publication

**Reviewer #6:**  Overall, this is an interesting review of an important area. However, several areas need attention and I do have a few issues to be addressed by the authors.

1. The meaning of the group was not indicated in figure legend of figure 1C.

2. The conclusion that the expression of Ptgds is upregulated in neomycin-associated injury seems not rigorous bescuse the authors did not verify the result of RNA.

3. The quality of the picture is not high. For example, in figure 3A, the merge of tunel and dapi should be overlapped.

4. Carefully check and improve the English writing in the manuscript.

7. PLOS authors have the option to publish the peer review history of their article (what does this mean? ). If published, this will include your full peer review and any attached files.

**Do you want your identity to be public for this peer review?** For information about this choice, including consent withdrawal, please see our Privacy Policy .

Reviewer #2: No

Reviewer #6: No

---

## [Author Response · Author response to Decision Letter 3]

23 Nov 2024

Dear Reviewer,

We sincerely appreciate your valuable comments and constructive feedback on our manuscript. Your suggestions have been instrumental in improving the quality of our study. Below, we provide detailed responses and revisions to address the issues you raised:

Regarding the Legend of Figure 1C

Thank you for pointing out the lack of clarity in the figure legend regarding the experimental group descriptions. We have revised the legend for Figure 1C and included detailed explanations of each experimental group in the revised manuscript to ensure clarity and comprehensibility. We hope this modification makes this section more intuitive and accurate.

Regarding the Rigor of the Conclusion on Ptgds Upregulation

We completely agree We fully agree with your suggestion that further validation of RNA experimental results is critical for enhancing the rigor of our conclusions. In fact, we have already conducted qRT-PCR experiments to validate these results, and the corresponding data have been incorporated into the revised manuscript (Supplementary Figure 1). The data clearly demonstrate the significant upregulation of Ptgds expression in the utricle following neomycin-induced damage. Based on these findings, we have also revised the discussion section to ensure a more scientifically robust conclusion.

Regarding the Quality of Figure 3A

Thank you for highlighting the quality issues with Figure 3A. Upon careful reevaluation, we found that due to technical limitations, there are spatial discrepancies between the TUNEL and DAPI signals, which prevent complete overlap. Specifically, DAPI, as a nuclear marker, exhibits uniform distribution, whereas Myosin7a selectively marks the stereocilia and cytoplasmic regions of cochlear and vestibular hair cells, reflecting its high spatial specificity. Following neomycin-induced damage, partial injury or death of vestibular hair cells resulted in Myosin7a signals covering only a subset of the DAPI regions. These findings are technically reasonable and align with the functional and biological differences between these two markers.

Regarding the Improvement of English Writing

Thank you for pointing out areas in need of improvement in the English writing. We conducted a comprehensive review of the manuscript and engaged professional editing services to optimize the language. These efforts were aimed at ensuring clarity and professionalism in presenting our findings. We believe these improvements have significantly enhanced the overall quality of the manuscript.

We sincerely thank you for your valuable suggestions, which have greatly contributed to refining our work. We hope these revisions adequately address your concerns. Should you have any further questions or recommendations, we are happy to make additional adjustments.

Once again, thank you for your support and constructive feedback.

---

## [Decision Letter · Decision Letter 3]

13 Jan 2025

PONE-D-24-17432R3Ptgds downregulation protect Vestibular hair cells from aminoglycoside-induced vestibulotoxicityPLOS ONE

Dear Dr. Nie,

Thank you for submitting your manuscript to PLOS ONE. After careful consideration, we feel that it has merit but does not fully meet PLOS ONE’s publication criteria as it currently stands. Therefore, we invite you to submit a revised version of the manuscript that addresses the points raised by the Academic Editor, prior to final decision.

We look forward to receiving your revised manuscript.

Kind regards,

Miriam A. Hickey, PhD

Academic Editor

PLOS ONE

**Editor Comments:**

Prior to final decision, please now address these comments from the Academic Editor.

Table 1

Monitor spelling

Please have all genes formatted similarly - the convention being italics for non-human genes.

Please consult here: https://journals.plos.org/plosone/s/submission-guidelines

Use correct and established nomenclature wherever possible.

For all imaging, please provide resolution i.e., actual size of each pixel, with size of z-step, and objective (cellular imaging, immunofluorescence sections).

Immunofluorescence section, please provide RRID number for antibody, if possible.

Please provide numerator, denominator and F values for all ANOVAs (Figure 1B, C, Figure 2 B, C D, Figure 3B, Figure 4 B, C, D and E).

Figure 1C: Please clarify whether cells with Ptgds knockdown were also treated with neo - this is not apparent in figure legend, but text suggests this is the case.

Please provide materials and methods for CCK8 method quoted in figure legends.

What are HEI-Oc1 cells (no information is apparent in materials and methods).

Please clarify/explain the reason for the much lower concentration of neo used for TUNEL experiments (3mM appears to have been used: "samples were treated for 12 h with neomycin (3 mM).").

Figure 3C: This must be analysed with at least 2-way ANOVA as there are two factors (treatments and gene). Please provide numerator, denominator and F value.

Discussion: Please discuss the relevance of the concentrations of neomycin used relative to concentrations observed in clinical samples.

Reviewers' comments:

Reviewer's Responses to Questions

**Comments to the Author**

1. If the authors have adequately addressed your comments raised in a previous round of review and you feel that this manuscript is now acceptable for publication, you may indicate that here to bypass the “Comments to the Author” section, enter your conflict of interest statement in the “Confidential to Editor” section, and submit your "Accept" recommendation.

Reviewer #2: All comments have been addressed

2. Is the manuscript technically sound, and do the data support the conclusions?

Reviewer #2: Yes

3. Has the statistical analysis been performed appropriately and rigorously? 

Reviewer #2: Yes

4. Have the authors made all data underlying the findings in their manuscript fully available?

Reviewer #2: Yes

5. Is the manuscript presented in an intelligible fashion and written in standard English?

Reviewer #2: Yes

6. Review Comments to the Author

Reviewer #2: The authors have adequately addressed your comments raised in a previous round of review. We feel that this manuscript should been now acceptable for publication.

7. PLOS authors have the option to publish the peer review history of their article (what does this mean? ). If published, this will include your full peer review and any attached files.

**Do you want your identity to be public for this peer review?** For information about this choice, including consent withdrawal, please see our Privacy Policy .

Reviewer #2: No

---

## [Author Response · Author response to Decision Letter 4]

8 Feb 2025

We sincerely appreciate your valuable comments and constructive feedback on our manuscript. Your suggestions have been instrumental in improving the quality of our study. Below, we provide detailed responses and revisions to address the issues you raised:

1. All typographical errors in Table 1 have been corrected. In accordance with the nomenclature conventions outlined in the PLOS ONE submission guidelines, non-human gene names have been italicized, while human gene names are presented in uppercase.

2. The Materials and Methods section has been expanded to include the following details: “After immunofluorescence staining, images were captured using a Zeiss LSM 800 confocal scanning microscope. Excitation light at two different wavelengths, 488 nm and 750 nm, was employed. The image resolution was set to 1024×1024 pixels, utilizing 20× and 63× magnifications, and the scanning speed was adjusted to 8 (with modifications made as needed based on image quality). The utricle was divided into the striolar region and the extrastriolar region. In each utricle, random 100 μm × 100 μm areas were selected from both the striolar and extrastriolar regions for imaging. The number of myosin7a-positive cells was manually counted using ImageJ software (NIH, USA). CellROX-stained images were acquired at a resolution of 1024×1024 pixels under 20× magnification using a REVOLVE FL microscope (Discover Echo, USA).”

3. The data related to the numerator, denominator, and F-values for all ANOVA analyses (Figure 1B, C; Figure 2B, C, D; Figure 3B; Figure 4B, C, D, and E) have been transferred to the supplementary data section.

4. We are grateful for your insightful comments. The corresponding clarifications have been integrated into the manuscript:“Compared to the negative control group, the expression of Ptgds in HEI-OC1 cells was significantly suppressed by approximately 70% after 48 hours of siRNA transfection.”

5. A comprehensive description of the CCK-8 assay methodology has been incorporated into the Materials and Methods section: “The CCK-8 (Cell Counting Kit-8) assay is utilized to evaluate cell viability or apoptosis. Initially, 1.2 × 10⁴ cells per well were seeded into a 96-well plate and allowed to adhere for 12 hours. Experimental groups were then established, encompassing a neomycin concentration gradient (0–50 mM, with a minimum of three replicates per group), a control group, a neomycin-treated group, and a neomycin-treated + siRNA group. Following 48 hours of transfection, neomycin was administered. After 24 hours of neomycin exposure, a 10:1 mixture of DMEM and CCK-8 solution was prepared, with 100 μL added to each well, followed by incubation at 37°C in the dark for 2 hours. Finally, absorbance at 450 nm was measured using a microplate reader, and cell viability (%) was calculated as (ODexperimental well – ODblank well) / (ODcontrol well – ODblank well) × 100%.”

6. A description of the HEI-OC1 cell line has been added to the Materials and Methods section: "HEI-OC1 cells are an immortalized mouse cochlear cell line widely utilized in research on ototoxicity and auditory-related mechanisms."

7. Due to the significantly lower number of hair cells in the utricle compared to HEI-OC1 cells cultured in 96-well plates, it was necessary to proportionally reduce the neomycin concentration and exposure time for neonatal mouse utricle damage. Based on a review of the relevant literature and experimental findings from our team, 3 mM neomycin exposure for 12 hours was determined to be the optimal concentration and duration for inducing approximately 50% utricle damage. The reduced neomycin concentration (3 mM) employed in the TUNEL assay was specifically chosen to induce apoptosis without causing excessive cell death, thereby facilitating clearer detection of apoptotic cells.

8. We appreciate your comment regarding the statistical analysis. We would like to clarify that only one independent variable (treatment) was involved in this analysis. Although the data from the three genes were presented together in a single graph for visualization purposes, each gene was analyzed separately using one-way ANOVA. Therefore, a two-way ANOVA is not applicable in this context. The detailed statistical results have been provided in the supplementary data file.

9. The Discussion section has been expanded to elucidate the correlation between the neomycin concentrations used in this study and those encountered in clinical practice: “Aminoglycosides (AGs) are a class of antibiotics widely used in clinical practice, known not only for their ototoxicity but also for their vestibulotoxicity. During clinical administration, AGs frequently cause damage to vestibular hair cells, which can lead to vestibular dysfunction. This dysfunction manifests as clinical symptoms such as ataxia, postural instability, dizziness, and vomiting. Neomycin, one of the commonly used aminoglycosides, is typically administered topically or orally in clinical settings. These routes of administration result in relatively low plasma concentrations of neomycin, thereby minimizing systemic exposure. However, despite the lower systemic absorption, neomycin is still associated with significant nephrotoxicity and ototoxicity. Due to these adverse effects, intravenous administration of neomycin is rarely employed in clinical practice [2].

While neomycin demonstrates favorable therapeutic efficacy, its clinical application is severely limited by its unavoidable side effects, particularly its toxicity to the auditory and vestibular systems. In this study, the concentration of neomycin used exceeds the typical plasma concentrations observed during clinical use. However, it aligns with concentrations commonly employed in in vitro studies designed to model ototoxicity. This approach enables a more accurate investigation of the mechanisms underlying aminoglycoside-induced vestibular damage, offering valuable insights that may contribute to the development of strategies aimed at mitigating these adverse effects in clinical practice.”

---

## [Editor Report · Decision Letter 4]

12 Feb 2025

PONE-D-24-17432R4Ptgds downregulation protect Vestibular hair cells from aminoglycoside-induced vestibulotoxicityPLOS ONE

Dear Dr. Nie,

Thank you for submitting your manuscript to PLOS ONE. After careful consideration, we feel that it has merit but does not fully meet PLOS ONE’s publication criteria as it currently stands. Therefore, we invite you to submit a revised version of the manuscript that addresses the points raised during the review process.

Unfortunately, the statistics (F values, numerator, denominator) are not apparent within the submitted supplementary file, which appears to contain Table S1 and Supplementary Figure 1, only.

Therefore, we invite you to submit a revised version of the manuscript that addresses this point.

A rebuttal letter that responds to the points above, from the academic editor. You should upload this letter as a separate file labeled 'Response to Editor'.A marked-up copy of your manuscript that highlights changes made to the original version. You should upload this as a separate file labeled 'Revised Manuscript with Track Changes'.An unmarked version of your revised paper without tracked changes. You should upload this as a separate file labeled 'Manuscript'.

We look forward to receiving your revised manuscript.

Kind regards,

Miriam Ann Hickey, PhD

Academic Editor

PLOS ONE
---

## [Author Response · Author response to Decision Letter 5]

20 Feb 2025

The statistics (F values, numerator, denominator) are not apparent within the submitted supplementary file, which appears to contain Table S1 and Supplementary Figure 1, only.

Response: We have modified the Table S1 and Supplementary Figure S1 as suggested, as shown in the revised Supporting Information.

---

## [Editor Report · Decision Letter 5]

23 Feb 2025

Ptgds downregulation protect Vestibular hair cells from aminoglycoside-induced vestibulotoxicity

PONE-D-24-17432R5

Dear Dr. Nie,

We’re pleased to inform you that your manuscript has been judged scientifically suitable for publication and will be formally accepted for publication once it meets all outstanding technical requirements.

Kind regards,

Miriam Ann Hickey, PhD

Academic Editor

PLOS ONE

Additional Editor Comments (optional):

All comments are now addressed.
---

## [Editor Report · Acceptance letter]

PONE-D-24-17432R5

PLOS ONE

Dear Dr. Nie,

I'm pleased to inform you that your manuscript has been deemed suitable for publication in PLOS ONE. Congratulations! Your manuscript is now being handed over to our production team.

Kind regards,

on behalf of

Dr. Miriam Ann Hickey

Academic Editor

PLOS ONE